# Netrin-1: A Modulator of Macrophage Driven Acute and Chronic Inflammation

**DOI:** 10.3390/ijms23010275

**Published:** 2021-12-27

**Authors:** Laura Ziegon, Martin Schlegel

**Affiliations:** Department of Anesthesiology and Intensive Care Medicine, Klinikum Rechts der Isar, Technical University Munich, 81675 Munich, Germany; laura.ziegon@tum.de

**Keywords:** netrin-1, neuronal guidance protein, chronic inflammation, acute inflammation

## Abstract

Netrins belong to the family of laminin-like secreted proteins, which guide axonal migration and neuronal growth in the developing central nervous system. Over the last 20 years, it has been established that netrin-1 acts as a chemoattractive or chemorepulsive cue in diverse biological processes far beyond neuronal development. Netrin-1 has been shown to play a central role in cell adhesion, cell migration, proliferation, and cell survival in neuronal and non-neuronal tissue. In this context, netrin-1 was found to orchestrate organogenesis, angiogenesis, tumorigenesis, and inflammation. In inflammation, as in neuronal development, netrin-1 plays a dichotomous role directing the migration of leukocytes, especially monocytes in the inflamed tissue. Monocyte-derived macrophages have long been known for a similar dual role in inflammation. In response to pathogen-induced acute injury, monocytes are rapidly recruited to damaged tissue as the first line of immune defense to phagocyte pathogens, present antigens to initiate the adaptive immune response, and promote wound healing in the resolution phase. On the other hand, dysregulated macrophages with impaired phagocytosis and egress capacity accumulate in chronic inflammation sites and foster the maintenance—and even the progression—of chronic inflammation. In this review article, we will highlight the dichotomous roles of netrin-1 and its impact on acute and chronic inflammation.

## 1. Netrin-1 Acts as a Chemoattractant and Chemorepellent Guidance Cue

The name netrin originates from the Sanskrit word netr, which means “the one who guides.” This reflects the fact that these laminin-like secreted neuronal guidance proteins were initially described as guidance cues in the developing central nervous system, where they act either as chemoattractive or as chemorepulsive cues. The netrin family consists of netrin-1, netrin-2, and netrin-3, and more distantly related netrin-4, netrin-G1, and G2 [1,2], which all belong to the superfamily of laminins [3]. Netrins are characterized by a preserved structure containing N-terminal domain laminin homologous domains, followed by three epidermal growth factor (EGF)-like repeats (LE) and a small, less well conserved, positively charged domain C-terminal [4,5]. 

In the early 1990s, netrin-1 was first described in the developing central nervous system [6,7]. Netrin-1 was shown to coordinate cell migration and axonal growth as a chemoattractant or chemorepellent, depending on the receptors expressed by the target cells. The chemorepulsion mediated by netrin-1 is facilitated by an Unc5 homodimer within close proximity and an Unc5/DCC heterodimer at longer ranges [8].

DCC has been further shown to signal netrin-1 dimerization with neogenin, another neuronal guidance receptor [6,8]. Moreover, it is suggested that the composition of the ligand receptor complex not only results in unique signaling mechanisms, but the different architectures of the ligand–receptor complex can also result in diverse signaling responses [6]. In addition to this complexity, it was recently shown that the post-transcriptional splicing of netrin-1 can significantly impact its biological functions [7].

Netrin-1 and its receptor DCC play an important role in different cellular processes such as cell adhesion, mortality, proliferation, cell survival, tissue organization, and cancer [9].

Moreover, the netrin-1 receptors Unc5b and DCC are so-called dependence receptors. These dependence receptors induce a caspase-triggered apoptosis in the absence of their ligand netrin-1 [9,10,11]. The importance of netrin-1 was shown in human tumors, such as brain tumors, neuroblastomas, prostate cancer, pancreatic cancer, and colorectal cancer. In these tumors, the gain of netrin-1 or the deletion of Unc5b and DCC prevents tumor cells from undergoing apoptosis and thereby promotes tumor growth and metastases [12].

Netrin-1 is also involved in the purinergic pathway, as it binds to the adenosine receptor Adora2b (A2BAR) [13,14,15]. By binding to A2BAR, netrin-1 increases intracellular cAMP levels, which can enhance the endothelial barrier function [13] and macrophage polarization [16]. However, it is of importance that the overall consequences of elevated cAMP levels are opposing in different cell types and organs [17,18]. Moreover, netrin-1 was shown to mediate epithelial cell adhesion and migration through the integrins α6β4 and α3β1 [19].

In 2005, Ly et al. [20] provided the earliest evidence that, outside the central nervous system, netrin-1 can also modulate the migration of granulocytes, monocytes, and leukocytes in and outside the inflamed tissue. Later it was shown that netrin-1 further promotes macrophage differentiation to an alternative activated phenotype [21].

Over the 30 years since its discovery, it has been well established that netrin-1 is involved in diverse biological processes far beyond the neuronal axon guidance. In this context, netrin-1 was found to play an important role in organogenesis [22], angiogenesis [23], tumorigenesis [11,24], promoting epithelium repair after injury [25], and inflammation. Given the numerous receptors and dichotomous signaling of netrin-1, it is not surprising that netrin-1 plays divergent roles in various settings of acute and chronic inflammation.

## 2. Acute Inflammation

Acute inflammation is a vital early physiological response to infection, trauma, injury, and a variety of other immunological challenges. The acute inflammatory response begins with an influx of neutrophils into the infected tissue to fight the infection and to create an inflammatory milieu by the release of granules and proinflammatory cytokines [26]. Subsequently, monocytes are attracted into this inflammatory milieu and support the neutrophils challenging the invading pathogen [27]. This initial phase of acute inflammation needs to be contained as the pathogen is cleared. In the resolution phase, anti-inflammatory and pro-resolving programs are initiated to control the acute inflammatory response and actively attain homeostasis. The impairment or absence of resolution leads to an overwhelming uncontrolled inflammatory response, resulting in chronic inflammation, persistent organ dysfunction, and even death [27]. In contrast to the numerous positive guidance cues controlling the influx of leukocytes into the inflamed tissue [26], negative guidance cues limiting this influx remained unrevealed for a long time [27]. Ongoing research has shown that netrin-1 can actively attenuate the acute inflammatory response and accelerate pro-resolving mechanisms, which cease the inflammation and help the inflamed tissue to return to homeostasis [27,28] (Figure 1 and Table 1). 

### 2.1. Acute Lung Injury

Acute lung injury (ALI) and the acute respiratory distress syndrome (ARDS) show, despite the progress of modern intensive care medicine, an incidence of up to 86.2 cases per 100,000 persons per year [47] and still a very high mortality (34.9–46.1%) [48] due to the lack of specific therapeutic options. The acute phase injury results from alveolar-endothelial injury and the formation of pulmonary edema [49,50]. ALI is driven by a neutrophil and monocyte influx into the alveoli. Several studies have shown that the netrin-1 level is decreased in ALI and inversely correlates with the cytokine levels and leukocyte infiltration into the lung [20,29,30]. The systemic intraperitoneal administration of netrin-1 mitigates lymphocyte trafficking into the inflamed lung, dampens the systemic inflammation, and diminishes the resulting lung injury [29,30]. The beneficial netrin-1 effects in ALI are mediated partly by Unc5b [20] as well as by A2BAR [29]. 

Mirakaj et al. [20] identified that the netrin-1 repression upon inflammation is mediated by NF-κB binding to a putative netrin-1 promoter region 69 bp upstream of its first codon [29]. Moreover, in tumorigenesis [51], by activation through free fatty acids [52], and hypoxemia [13] it was shown that netrin-1 is induced via direct binding of NF-κB subunit p65 (RelA) to a netrin-1 promoter [33]. However, the mechanisms by which netrin-1 is activated or repressed in other inflammatory conditions need further study.

In ALI the A2BAR binding of netrin-1 enhances cAMP, which consequently increases epithelial sodium channel-mediated alveolar fluid clearance [32]. This effect was reduced by the specific inhibitor PSB1115 of A2BAR. Lung ischemia reperfusion injury (LIRI) leads to an increase of T cells and macrophages and a decrease of netrin-1 and Treg cells in vitro and in vivo. The positive correlation between netrin-1 and the upregulated Treg cell population, which reduces LIRI, is also mediated by the A2BAR receptor [31]. 

Both the netrin-1 receptor A2BAR and the UNC5B receptor affect ALI. Ko et al. [33] established a hypobaric hypoxia-induced lung injury model that showed a decrease in netrin-1 and its receptor UNC5HB. This decrease was prevented by pre-treatment with netrin-1. 

In a sepsis rat model, levels of netrin-1 and its receptor UNC5B were downregulated. The transfection of pcDNA-netrin-1 increased netrin-1 and UNC5B expression, which resulted in a prevention of lung injury caused by sepsis [53]. 

In another study, Berg et al. [54] investigated the role of netrin-1 during endotoxin-induced lung injury and found the importance of hypoxia-inducible factor HIF-1α for the induction of myeloid-derived netrin-1, which led to a decrease in the NK cell number, examined by the deletion of netrin-1 in the myeloid compartment. These findings emphasize the regulation of lung inflammation by myeloid-derived netrin-1 through the modulation of NK cell infiltration. 

### 2.2. Acute Abdominal Inflammation

Acute abdominal inflammation includes a broad spectrum and a variety of very divergent organs, causing inflammation in the abdominal cavity. Despite the wide variety of abdominal inflammations, the prophylactic and therapeutic application of netrin-1 dampens the inflammation and improves the outcome. However, netrin-1 expression in the abdomen was found to be both induced or repressed, depending on the stimulus and the organ.

#### 2.2.1. Inflammatory Bowel Diseases

Inflammatory bowel diseases (IBDs) are a class of heterogeneous inflammatory conditions of the small intestine and the colon. The two major forms of IBD are Crohn’s disease (CD) and ulcerative colitis (UC) [55]. Both are characterized by relapsing and remitting episodes of acute inflammation in mucosal lining and are characterized by intestinal barrier dysfunction and neutrophil influx. A significant expansion of tissue-resident and bone-marrow-derived macrophages in the lamina propria, in CD, and in the muscularis mucosa have been reported [56,57]. The local imbalance of pro- and anti-inflammatory cytokines [58] leads to an increase in both M1-like proinflammatory macrophages [59] as well as CD163+ M2-like pro-healing macrophages [60,61].

In acute experimental colitis, a model for inflammatory bowel disease (IBD), netrin-1 is induced in the mucosal lining of the whole colon. Aherne et al. [34] found that intrinsic netrin-1 released by the intestinal epithelium, as well as supplemented netrin-1, limits the neutrophil, but not the monocyte, influx into the mucosa, thereby attenuating mucosal inflammation, while the epithelial barrier function remains unchanged [34,62]. Mice partially deficient in netrin-1 displayed an aggravated inflammatory response. Those beneficial effects of netrin-1 application vanished in mice deficient in the adenosine receptor Adora2b (A2BAR) [34]. This indicates that the purinergic signaling pathway contributes to the favorable effects of netrin-1 in inflammatory bowel disease. However, the effects of netrin-1 on the mucosal macrophage expansion and differentiation have yet to be studied. 

IBD confers an increased risk of colorectal cancer. The transformation of colorectal cancer that could arise from IBD with a higher risk is mediated by the NF-κB signaling pathway [11,63]. Paradisi et al. [63] showed a selected upregulation of netrin-1 and its receptors DCC and UNC5H in IBD patients and concluded that this upregulation is causal for the development of colorectal cancer.

#### 2.2.2. Acute Pancreatitis

Acute pancreatitis is an acute inflammation of the pancreas induced by diverse etiological factors (such as ethanol, duct obstruction, trauma, metabolic disorder, and many more) [64]. 

The premature activation of digestive enzymes (e.g., trypsin, elastase, and lipase) induces an autodigestion of the pancreas causing an overwhelming local and systemic inflammatory response. This systemic inflammatory response in severe acute pancreatitis is associated with a 10–30% mortality. The acute response is mainly driven by an infiltration of neutrophils and monocytes into the pancreas, as well as by an activation of Kupffer cells, peritoneal and alveolar macrophages (as reviewed in [65,66]).

In acute pancreatitis, netrin-1 protein levels are increased within the pancreas [36]. Prophylactic treatment with intravenously administered netrin-1 diminishes the systemic inflammatory cytokines and the neutrophil degranulation in the pancreas as well as in the lung [36]. As macrophages are key players in acute pancreatitis, we suggest an effect of netrin-1 on macrophage dynamics and polarization. However, the receptors, the signaling pathways mediating netrin-1′s beneficial effects, and the effect of netrin-1 on macrophages in acute pancreatitis are still undetermined.

#### 2.2.3. Acute Peritonitis

Acute peritonitis is a final common path of a variety of intra-abdominal causes commonly associated with a perforation of the abdominal wall. Especially in intensive care units (ICUs), acute peritonitis is a relevant problem due to its high incidence [67], high mortality rate [68,69], and limited therapeutic options. Acute peritonitis is characterized by the typical sequence of acute inflammation, starting with edema and followed by an influx of neutrophils and monocytes. 

In a model of Zymosan-A-induced peritonitis, netrin-1 levels decrease in the peritoneal cavity during the inflammatory response [35]. The therapeutic intravenous reconstitution of netrin-1 diminishes the acute inflammatory response by limiting the neutrophil influx into the peritoneum. These protective netrin-1 effects are blunted in A2BAR KO-mice. In a subsequent study, Mirakaj et al. [27] showed that netrin-1 not only has an anti-inflammatory effect but furthermore that it can promote the resolution of acute inflammation and act synergistically with the specialized pro-resolving lipid mediator resolvin D2 (RvD2). Netrin-1 accelerates the return to homeostasis by limiting the influx of neutrophils into the inflamed tissue, enhancing the synthesis of pro-resolving lipid mediators and stimulating monocyte efferocytosis [27]. This study further focused on the vagal nerve stimulating netrin-1 expression at homeostasis and in inflammation. This was the earliest evidence that the neuronal inflammatory reflex can actively modulate the expression of netrin-1 and that netrin-1 can effectively promote resolution of acute inflammation.

### 2.3. Hypoxia and Ischemia–Reperfusion Injury

Hypoxia can be both a cause and a result of acute inflammation. That hypoxia can induce inflammation was first noticed by the finding that patients with high altitude illness show increased levels of proinflammatory cytokines, as well as pulmonary and cerebral vascular leakages [70]. These observations were supported by animal experiments in which short-term exposure to hypoxia induced similar patterns of proinflammatory cytokines, vascular leakage, and leukocyte trafficking into the hypoxic organs [13,71]. In acute inflammation, however, hypoxia occurs due to an increased oxygen demand in the inflamed tissue and often is aggravated by a reduced oxygen supply. This hypoxic milieu further worsens the local inflammation. The activation of hypoxia inducible factor (Hif)-1α and Hif2α promotes macrophage migration, phagocytosis, and cytokine production and fosters the local inflammatory environment [72,73].

The relationship and the pathophysiological interactions between hypoxia and inflammation have been discussed in detail in recent reviews [74,75]. Often at this primary hypoxic injury, the hypoxic induction of inflammation is followed by a secondary reperfusion injury aggravating the hypoxic injury [76]. Reperfusion injuries are clinically critical in a variety of conditions such as hemorrhage, sepsis, revascularized myocardial infarction, and organ transplantations. 

#### 2.3.1. Hypoxia 

As discussed earlier, hypoxia is a potent inductor for the expression of netrin-1 [13,77]. Rosenberger et al. [13] showed that this netrin-1 induction in hypoxia is promoted by HIF-1α binding to a hypoxia responsive element (HRE) on the netrin-1 promoter region to control the hypoxic inflammation. This study also showed that netrin-1 is efficiently able to attenuate the inflammation and neutrophil influx elicited by hypoxia in the lung, colon, and kidneys [13].

In this hypoxia study, netrin-1 binding to Adora2b but not to Unc5b increased intracellular cAMP levels. These increased cAMP levels enhanced the endothelial barrier function [13]. Furthermore, increased intracellular cAMP levels in macrophages have been shown to induce a macrophage phenotype switch from proinflammatory M1 macrophages to pro-resolving M2-like macrophages [78]. In line with these findings, kidneys and spleen overexpressing netrin-1 show a phenotypic shift from M1 macrophages to a M2-like phenotype by the induction of PPARγ pathways [16]. A2BAR-deficient mice, on the other hand, showed an impaired polarization towards M2 macrophages in inflammation [79]. Taken together, even though netrin-1 induces intracellular cAMP levels, the overall consequences are opposing in different cell types and organs, depending on the target mediating the cAMP effects [17,18]. 

Jin et al. [80] demonstrated the importance of netrin-1 in tumor development under hypoxia, which is a microenvironment condition of non-small cell lung cancer by the detection of an epithelial-to-mesenchymal transition, which is mediated by netrin-1 through the phosphoinositide 3 kinase/AKT pathway. 

#### 2.3.2. Acute Ischemic Kidney Injury

Ischemic renal failure is a classical acute inflammation driven by leukocyte influx into the kidney, vascular leakage, proinflammatory cytokine release, and apoptosis [81]. Several studies have shown that activated proinflammatory macrophages accumulate within the first 24 h after kidney I/R. Depletion of the macrophages prior to the kidney ischemia leads to a diminished kidney injury. However, a depletion of macrophages later during the resolution phase impaired the tissue regeneration. Netrin-1 is highly induced in tubular epithelial cells [38,39] and urinary excretion [38,39,40] following kidney IR.

Netrin-1 regulates immune cell migration, cytokine production, like IL-6, IL-1β, and TNF-α, and macrophage polarization [82]. UNC5B, which was detected on immune cell surfaces, is expressed in proximal tubular epithelial cells and vascular endothelial cells [82]. Neogenin is expressed in all segments of the nephron in the basolateral surface [82]. Anti-inflammatory effects in acute kidney injury (AKI) are mediated via the netrin-1 receptor UNC5B by the downregulation of inflammatory cytokines [40,82]. Netrin-1 was upregulated at the translational level in tubular epithelial cells 3 h after reperfusion in I/R injury. A shift from endothelial cells that show lower netrin-1 levels after hypoxia to epithelial cells and a downregulation of netrin-1 in peritubular cells was detected [82].

The increased protein levels in the renal tubular epithelium are induced by an activation of ERK and AKT pathways [83]. Despite the abundant netrin-1 expression in the tubular epithelial cells following kidney IR, the additional systemic administration of netrin-1 [37,39,40] and the renal overexpression [38] of netrin-1 attenuate systemic inflammation, neutrophil and monocyte influx, and renal failure. The renal overexpression of netrin-1 prevents ischemia–reperfusion-induced apoptosis and even induces tubular epithelial proliferation, in addition to limiting the leukocyte migration and the acute inflammatory response [39]. In kidney IR netrin-1, beneficial effects can be reversed by a functional blockade of a Unc5b [39]. The deletion of UNC5B reduced netrin-1-mediated protective effects and exacerbated AKI [82,84]. The signaling pathway through NF-κB inhibits COX-2 and PGE2 expression, which mediates stimulation of immune cell function, chemotaxis, and chemokine expression [82]. Netrin-1 regulates macrophage polarization to the M2 phenotype, which produces anti-inflammatory molecules such as IL-10 or TGF-β1 through the PPAR pathway [82]. While IFN-γ induces M1 markers, treatment with netrin-1 suppresses expression and enhances development of the M2 phenotype with anti-inflammatory effects [82]. Netrin-1 reduced inflammation, neutrophil invasion, and cell apoptosis, resulting in improved kidney function [16]. The value of netrin-1, even as a predictive marker of AKI, was shown in a prospective study on patients undergoing cardiac surgery, in which the urinary netrin-1 levels 6 h after a cardiopulmonary bypass highly correlated with the duration and severity of the developing AKI as well as the length of hospital stay [85]. Even though netrin-1 is a promising biomarker and therapeutic target, the mechanisms by which netrin-1 protects the kidney, how it impacts macrophage dynamics and polarization, and its roles in the different phases of kidney injury need to be further investigated. 

#### 2.3.3. Liver Ischemia and Reperfusion Injury

Liver ischemia–reperfusion injury is a common clinical picture seen in liver surgery, trauma, succeeding hemorrhagic shock with fluid resuscitation, and sepsis. These result in liver failure, remote organ failure, and systemic inflammatory response syndrome (SIRS) [86]. The ischemia–reperfusion injury can be divided into two distinct phases, as described by Jaescke et al. [87]. The initial phase is dominated by Kupffer cell activation, release of reactive oxygen species, and systemic neutrophil activation [88]. The second phase is dominated by neutrophil influx into the liver causing hepatocyte damage and cell death [88]. In contrast to the kidney, the netrin-1 expression in the liver is diminished following liver ischemia–reperfusion (LIR). Netrin-1 administration, however, attenuates LIR by limiting neutrophil influx and intrahepatic inflammation. More important, netrin-1 even enhances the resolution of the inflammation and liver regeneration following LIR [28]. Netrin-1 induces pro-resolving lipid mediators and promotes the polarization of monocytes into pro-resolving, anti-inflammatory macrophages inducing the resolution. These pro-resolving macrophages and tissue-resident Kupffer cells show an increased efferocytosis and impact tissue repair by the induction of liver growth factors. These beneficial effects of netrin-1 in liver ischemia–reperfusion injury are adenosine A2B receptor-dependent [28]. 

#### 2.3.4. Acute Myocardial Infarction and Reperfusion Injury

Acute occlusion of a coronary artery, acute myocardial infarction (MI), is a leading cause of morbidity and mortality worldwide. In contrast to atherosclerosis, which is a chronic low-grade inflammation as discussed below, myocardial infarction is an acute inflammation. The blockage of a coronary vessel leads to acute myocardial hypoxia and cell death. Following the onset of an acute MI, the standard treatment of choice is to restore blood flow and oxygen supply as early as possible using primary percutaneous coronary intervention (PPCI) or thrombolytic therapy if PPCIs is delayed [89]. Even though the reperfusion is vital to salvage cardiomyocytes from cell death, the reperfusion itself causes a secondary inflammatory response driven by a release of chemoattractants, ROS, inflammatory cytokines, and neutrophil influx [76,90]. The reperfusion induces adhesion molecules on the endothelium to direct neutrophils and proinflammatory Ly6C^hi^ monocytes into the ischemic cardiac muscle [91]. In this critical reperfusion phase, netrin-1 is downregulated [44]. In an ex vivo myocardial ischemia model on the Langendorff perfusion, netrin-1 proved to be cardioprotective by reducing both hypoxia injury and reperfusion injury. Preconditioning with netrin-1 further reduces the myocardial infarct size and cardiomyocyte apoptosis through NO and ERK1/2 upregulation [41]. The pharmacological blockade of DCC prior to treatment abrogated the netrin-1-mediated cardioprotection. Preconditioning and postconditioning hearts with netrin-1 prevents intracardial superoxide and H_2_O_2_ production through the phosphorylation of eNOS and ERK1/2. In that, netrin-1 preserves mitochondrial function in cardiomyocytes and endothelial cells following myocardial infarction [42,43,46]. These cardioprotective netrin-1 effects are dose-dependent and only seen at very distinct netrin-1 concentrations, from 5 to 10 µg/kg [43]. DCC deficiency in mice or the pharmacological blockade of ERK1/2 and NO activation abolished the cardioprotective netrin-1 effects [42]. Daliang et al. [92] found a decrease of serum netrin-1, myocardial netrin-1, and DCC expression in an AMI model, but an increase after aerobic exercise. This could reduce myocardial fibrosis, and indicates a role for netrin-1 and its receptor DCC during AMI. 

In a murine heart transplant model, netrin-1 administration before the reperfusion ameliorated the reperfusion injury. Netrin-1 improved the myocardial function, decreased cardiomyocyte apoptosis, and decreased neutrophil and monocyte influx into the heart [44]. Further, netrin-1 induces the polarization of alternative activated M2 macrophages (AAM) in injured hearts [44] by the activation of the PPARγ pathways. While the netrin-1 releasing cells were not studied in those studies mentioned, it was shown by Ke et al. [45] that the cardioprotective effects of netrin-1 can be achieved by the intramyocardial injection of netrin-1-transfected mesenchymal stem cells (MSCs) after coronary artery ligation. The application of netrin-1 MSCs reduces the infarct size and prevents hypertrophic cardiac remodeling by the promotion of NO liberation and neoangiogenesis [45]. Altogether, these studies suggest that netrin-1 can mitigate the acute inflammatory response and accelerate wound healing following acute MI. However, the source of netrin-1 in healthy and hypoxic myocardial tissue remains an open question.

Patients with acute coronary syndrome (ACS) showed a high netrin-1 level on arrival at the emergency room, which decreased after angiography in patients with TIMI 3 flow (thrombolysis in MI; complete perfusion) [93]. High serum netrin-1 levels were found in elderly females with ACS and correlated with a bad outcome. Reasons for this could include more severe ACS and induced netrin-1 expression by hypoxia [94]. 

Li et al. [14] showed not only elevated netrin-1 in the blood of AMI patients, but also increased infarct size in mice with the deletion of netrin-1 in the myeloid compartment (Ntn1loxP/loxP Lyz2 Cre+ mice). The cardioprotection of PMN-dependent netrin-1, which is mainly synthesized by neutrophils, is mediated by the migration of neutrophils via an autocrine loop through myeloid adenosine A2b signaling [14]. These findings confirm a functional role of netrin-1 interaction with ADORA2B in attenuating hypoxia-driven inflammation in AMI [14,34].

## 3. Chronic Low-Grade Inflammation

In contrast to acute inflammation, chronic inflammation is a long-lasting, non-resolving adaptive immune response chiefly maintained by macrophages and lymphocytes. It is characterized by the simultaneous presence of active inflammation, tissue destruction, and tissue repair. Chronic inflammation can result from the host’s impairment of resolution programs following an acute inflammation. In these cases, the host defense is unable to effectively overcome the pathogen and therefore impairs the macrophage egress or apoptosis. Chronic inflammation can also develop as an independent low-grade inflammatory condition from the start, as in arthritis, chronic lung disease, obesity, and atherosclerosis. These ongoing chronic inflammatory processes involve tissue proliferation, ineffective apoptosis, fibrosis, and necrosis, which can result in deteriorated organ function or organ failure. Chronic inflammation is further linked to the development of a variety of cancers, such as bladder, cervical, gallbladder, liver, gastric, intestinal, and esophageal cancer (reviewed in [95]). In colorectal cancer, for example, the upregulation of netrin-1 and its receptor UNC5B correlates with the number of cancer-associated fibroblasts (CAFs). The inhibition of netrin-1 in colon cancer inhibits the increase of CAFs, resulting in decreased cancer stemness and plasticity [96]. Various murine models addressing the role and the expression of netrin-1 and its different receptors in chronic inflammation are summarized in Table 2.

### 3.1. Netrin-1 in Atherosclerosis

Atherosclerosis and cardiovascular disease (CVD) are the leading causes of death worldwide [107]. Most deaths attributed to CVD are due to myocardial infarction and stroke [108]. Atherosclerosis is a macrophage-driven, chronic, non-resolving inflammation [109]. Monocytes activated by lipoproteins infiltrate the arterial wall and progress to cholesterol-laden foam cells [110] (Figure 2). In a recent study, Van Gils and her colleagues [77,99] showed that these macrophages, in human and murine atherosclerotic plaques, secrete a high level of netrin-1 and thereby drive the progression of atherosclerosis. Bone marrow transplantation of Ntn1−/− cells into LDL−/− mice, which develop atherosclerosis on a high-fat diet, decreased plaque size development by 45%.

Netrin-1 produced by macrophages plays a proatherogenic role: first, it prevents macrophage migration and the egress from the plaque; second, it promotes lesion progression while enhancing the chemoattraction of smooth muscle cells and their recruitment into the intima [21,111,112]. 

The expression of netrin-1 and Unc5b in the atherosclerotic plaque are induced by hypoxia [77]. This hypoxic induction is dependent on HIF-1α, which itself is induced by hypoxic stressors such as oxLDL [113]. In the progression of the disease, these inflammatory cholesterol-laden foam cells persist in the plaque and contribute to local and systemic inflammation [111,114]. 

The continued monocyte recruitment, the expansion of lipid-laden macrophages in the lesion [115], and the impaired egress lead to the formation of large necrotic cores and a thinning of the fibrous cap [116]. Taken together, all these mechanisms contribute to a destabilization of the plaque, resulting in myocardial infarction [117]. It was shown that the deletion of netrin-1 in myeloid cells reduces atherosclerosis plaque size [111]. 

On the other hand, netrin-1 induced by TNF-α and produced by endothelial cells inhibits the entry of monocytes into the plaque by reducing the expression of adhesion molecules as well as the chemotaxis of leukocytes by the suppression of cytokine production, toll-like receptor (TLR) 4, and the NF-κB signaling pathway [21,111,112]. Stimulation with TNF-α in vitro showed a reduction in the expression of VCAM-1, ICAM-1, and E-selectin when cells were co-stimulated with netrin-1. They inhibit the expression of IL-6 and COX-2 [21], among other things. The downregulation of ICAM-1, IL-6, and MCP-1 that can also be induced by the UNC5B blockade was also observed in atherosclerosis patients [21]. These different effects may depend on the stage of atherosclerosis.

The importance and benefits of controlling systemic inflammation in atherosclerosis was shown in the CANTOS trial, a prospective patient study [118]. The anti-inflammatory treatment with an anti-IL-1β monoclonal antibody resulted in a decreased incidence of cardiovascular events and deaths without altering the cholesterol levels [118]. Under pro-atherogenic conditions, netrin-1 expression by endothelial cells is downregulated and thereby facilitates increased monocyte and macrophage influx into the arterial wall [119,120]. However, the plaque progression, which is driven by an on-going influx and accumulation of macrophages in the plaque, is not a one-way road. Fisher and Randolph [121] showed that plaque regression can be induced by facilitating macrophage egress from the arterial wall in a CCR7-dependent manner [122]. Netrin-1 blocks this directed macrophage migration towards the CCR7 ligands CCL-19 and CCL-21 and towards MCP-1 [99]. Netrin-1 disrupts the Rac1 signaling pathway, the actin cytoskeleton reorganization, and the cell polarization, thereby mitigating macrophage egress from the plaque [99]. These netrin-1-mediated chemotactic effects are inhibited by the blockade of Unc5b. 

In this context, Yang et al. [98] examined the expression of netrin-1 and its receptor UNC5B at the mRNA and protein level in Raw264.7 macrophages in response to ox-LDL. The data showed a downregulation of CCD7 that can bind the chemokines CCL19 and CCL21, an upregulation of UNC5B in an NF-κB-dependent manner, which inhibited foam cell migration [98]. However, more studies are needed to understand these mechanisms more thoroughly. 

In atherosclerosis, the bone-marrow transplantation of Ntn-1−/−-deficient cells in LDLR−/− mice leads to a significant increase in macrophage egress from the plaque [99]. In contrast to its effects on macrophages, netrin-1 recruits smooth muscle cells (SMC) into the plaque and promotes plaque growth [99]. The chemoattraction of SMC by netrin-1 is mediated through neogenin and not through Unc5b, which is responsible for the impaired egress of cholesterol-laden macrophages from the plaque. The restored macrophage migratory capacity in hematopoietic netrin-1-deficient mice led to a decrease in macrophages, T cells, and necrotic core area in the plaque [99]. Overall, the netrin-1 deletion in hematopoietic cells promotes a significant reduction in plaque size and complexity. Altogether this study demonstrates for the first time that netrin-1 secreted by plaque macrophages actively promotes the progression of atherosclerosis in an Unc5b-dependent manner.

Netrin-1, however, not only promotes the onset and propagation of atherosclerosis, but also hinders its regression. In established atherosclerosis, cholesterol lowering alone is insufficient to promote plaque regression, as monocyte-specific netrin-1 impairs the reorganization of the plaque’s immune cell landscape and hinders monocyte egress [100]. We were able to show that lipid lowering in established atherosclerosis and silencing macrophage-specific netrin-1 by a tamoxifen-inducible knock-out reduced the infiltration of intimal smooth muscle cells, macrophage accumulation, and survival in plaques, while markers of inflammation such as IL-10 and efferocytosis were simultaneously increased [100]. The atherosclerotic plaque burden in the thoracic aorta decreased by 50% after the deletion of monocyte-specific netrin-1, while it remained unchanged in littermate controls despite an effective lipid lowering. Therefore, targeting netrin-1 could be a promising approach also for already advanced atherosclerotic disease. 

Bruikman et al. [123] focused on netrin-1 levels in patients and found a negative correlation between netrin-1 and arterial wall inflammation, subclinical atherosclerosis, and plaque volume. However, there was no significant difference in netrin-1 plasma concentrations between patients with stable vs. unstable plaques. 

In another study, Bruikman and her colleagues [97] found a genomic variant of netrin-1 leading to a familiar premature atherosclerosis. The change in an arginine to leucine amino acid at position 1769 (NTN1 c.1769G > T; p.Arg590Leu) led to fewer anti-inflammatory effects on endothelial cells, and a strong blockage in macrophage emigration.

All these results indicate the important role of netrin-1 and its receptors in the process of atherosclerosis. However, given the dual function of netrin-1 in atherosclerosis, more insight is needed to determine at which stage of atherosclerosis netrin-1 supplementation or netrin-1 blocking substances will ameliorate atherosclerosis.

### 3.2. Aortic Abdominal Aneurysma

Aortic abdominal aneurysms (AAAs) are a potentially life-threatening dilatation of the abdominal aorta. An AAA is regarded as a long-term consequence of cardiovascular disease, even though its molecular mechanisms are not fully understood. More recent studies have revealed chronic inflammatory processes in the aortic vessel wall, including extracellular matrix (ECM) degeneration by matrix metalloproteinase (MMP) activation and oxidative stress.

A recent study identified, in damaged areas of thoracic aortic aneurysms, an upregulation of neuronal guidance proteins, including netrin-1 and its receptors UNC5B [124]. In line with these findings, Hadi et al. [125] describe how macrophage-derived netrin-1 promotes AAA progression through the release of MMP-3 from vascular smooth muscles by binding its receptor neogenin-1 [125]. Therefore, netrin-1 might be an important player in mediating the crossing between inflammatory processes and the erosion of the extracellular matrix in AAAs in both mice and humans [125].

### 3.3. Obesity 

The rates of obesity, particularly excessive visceral adiposity and its comorbidities, insulin resistance type 2 diabetes (T2D), and cardiovascular diseases, have increased dramatically over the past 25 years throughout the world [126,127] and are major threats to global health [128,129,130]. Chronic low-grade inflammation, mediated by the innate and adaptive immune system, has emerged as the key pathogenic link between obesity and its comorbidities [131].

It is now generally accepted that chronic low-grade inflammation in obesity is dominated by macrophages infiltrating the expanded adipose tissue in obese individuals [132,133,134,135]. These adipose tissue macrophages (ATMs) produce mainly proinflammatory “M1” cytokines such as TNF-α [136], interleukin (IL)-1β [137], and CCL2 [138], contributing to the persistence of local and systemic inflammation and increasing insulin resistance (IR) [52]. Recently, Ramkhelawon and her colleagues [52] showed that, in obesity, netrin-1 plays a pivotal role in the retention of ATMs in visceral adipose tissue (VAT) and promotes not only systemic inflammation but also metabolic dysfunction. This study showed that netrin-1 and its receptor Unc5b are highly upregulated in ATMs from obese compared to lean mice and humans. Other netrin-1 receptors, however, such as Neo1 and DCC, are expressed similarly in lean and obese individuals [52]. Free fatty acids, such as palmitate, can induce the expression of netrin-1 and Unc5b in BMDM directly in a NF-κB-dependent manner. Moreover, netrin-1 was shown to be induced by a paracrine release of cytokines, such as TNF-α and IL-6, but not IL-4, from adipocytes [52]. Murine 3T3-L1 adipocytes treated with free fatty acid stimulated BMDMs and conditioned media highly induces Ntn-1 and Unc5b expression. The blockade of TNF-α and IL-6 abrogates this netrin-1 and Unc5b induction [52]. 

In obesity, netrin-1 hinders ATM egress from the VAT, protects ATM from apoptosis, and induces an inflammatory M1-like phenotype. Netrin-1 further mediates chemostasis in an Unc5b-dependent manner and thereby actively fuels inflammation in the VAT. Moreover, netrin-1 impedes the migration of ATM egress from the VAT, as it mitigates migration towards CCL-19 [99]. This impairs the resolution of the ongoing chronic inflammation. WT mice were reconstituted with Ntn1−/− bone-marrow cells and fed a high-fat diet (HFD) to study the impact hematopoietic netrin-1 in vivo [101]. The netrin-1 deficiency did not protect the mice from gaining weight and becoming obese. However, the adipose tissue proved to be less inflamed, as fewer macrophages were found in the VAT. Bead-labeling studies showed that netrin-1 did not impact the recruitment of monocytes into the VAT but mitigated the egress from the VAT and thereby led to the increased ATM accumulation [101]. Functionally, the decreased local inflammation in mice with hematopoietic netrin-1 deficiency resulted in an improved glucose and insulin tolerance. We were able to show that macrophage-specific netrin-1 affects the metabolism in HFD-fed mice and leads to increased weight gain compared to Ntn1-deficient mice. Ntn1 further promotes macrophage differentiation, impairs lipid handling and migration, and promotes proinflammatory mediators [101].

Sanders et al. [139] showed in mouse experiments that expression of the netrin-1 receptors Unc5b and DCC is altered in the arcuate nucleus in fetuses developing in obese mothers [139] and that these alterations impact developmental pathways important for fetal development. These results may explain the defective neuropeptide Y innervation of the paraventricular nucleus of obese mothers [139].

A recent clinical study showed that circulating netrin-1 levels positively correlated with fasting glucose, the HbA1c level, and the insulin resistance index. This suggests that netrin-1 may be a sensitive indicator for early type 2 diabetes [140]. Together, these studies indicate that netrin-1 is a key player in obesity-induced inflammation and functionally impacts glucose tolerance.

### 3.4. Rheumatoid Arthritis and Osteolysis

Netrin-1 expression by osteoblasts and synovial fibroblasts is enhanced by IL-17. Binding its receptor UNC5B on the osteoclasts’ surfaces, netrin-1 initiates the activation of SHP1. This results in a blockade of the multinucleation, but not in the differentiation, of osteoclasts. Hence, netrin-1 prevents bone erosion in autoimmune arthritis and age-related bone destruction [104]. As netrin-1 is highly expressed in osteolysis, its blockade mitigates osteolysis by altering osteoclast differentiation [105]. In contrast, Mediero et al. [105] showed that anti-netrin-1 and anti-UNC5B antibodies increase bone density and reduce inflammatory infiltrates and particle-induced bone pitting. Netrin-1 also mediates osteoclastogenesis by autophagy in vitro and in vivo [106]. The netrin-1 blockade has been shown to reduce K/Bxn-induced arthritis [141]. 

Zhu et al. [142] found that osteoclast-secreted netrin-1 induces sensory nerve growth responsible for pain in osteoarthritis. The inhibition of netrin-1 or its receptor DCC was able to reduce the pain. Taken together, netrin-1 seems to differentially regulate osteoclastogenesis in different settings, and its therapeutic potential still needs to be determined.

### 3.5. Diabetes

A recent study found netrin-1 plasma levels to be decreased in patients with newly diagnosed diabetes mellitus type 2, and even suggest that netrin-1 should be assessed for its potential in early diabetes screening [143]. 

A major clinical challenge in diabetes is the diabetic peripheral neurovascular diseases (DPNVs), for which no effective treatment is currently available. In this framework Zhang et al. [102] showed that netrin-1 might mitigate DPNVs. In their study, transplanted adipose-derived stem cells, which have potential for differentiation and tissue restoration, were exposed to netrin-1 differentiated into endothelial cells and formed small capillaries [102]. Netrin-1 not only promotes neuronal migration but also regulates the survival, adhesion, migration, proliferation, and differentiation of endothelial cells. A negative effect on the proliferation and apoptosis of stem cells was shown in previous studies because of hypoglycemia. In this recent study [102], the important role of netrin-1 in proliferation, migration, adhesion, and apoptosis was shown under high-glucose conditions in vivo and in vitro. The combined signaling pathway AKT/PI3K/eNOS/P-38/NK-κB and the expression of many growth factors, such as VEGF, TNF-α, EGF, and netrin-1, were upregulated [102]. These results suggest that netrin-1 might be a potential target for DPNVs. 

On the other hand, Toque et al. [144] showed that netrin-1, not only reduces the high glucose induced by vascular endothelial dysfunction and limits the impairment of the nitric oxide synthase, but also suppresses inflammatory and apoptotic processes. 

Other studies have characterized the protective effect of netrin-1 on the apoptosis of β-cells [145], in diabetic retinopathy [146], diabetic nephropathy [147], and diabetes-associated myocardial infarction [45]. Goa et al. [148] reported a preventive role of netrin-1 against diabetes progression via dual action on islet insulin secretion and inflammation. Since diabetes and its co-morbidities are associated with hyperactive inflammation, netrin-1 could be a promising therapeutic candidate that reduces hypoglycemia while also ameliorating inflammation for diabetic individuals. 

Jiao et al. [103] showed that UNC5B is upregulated in diabetic kidney diseases (DKDs) and promoted by high glucose levels. Netrin-1 and its receptor UN5B are protective during DKDs because of the inhibition of netrin-1 by binding on UNC5B, which eliminates the negative effects of netrin-1 on tubulogenesis. 

The upregulation of the receptor UNC5B was examined in retinal microvessels, which enhances vascular integrity and is protective for adaptation to diabetes [7]. Elevated levels of UNC5B are associated with chronic inflammation such as atherosclerosis obesity, oxygen-induced retinopathy, and retinal vessel sprout [7].

Miloudi et al. [7] showed that metaloproteinase-9 cleaves netrin-1 into fragments. These fragments of truncated netrin-1 promote diabetic retinopathy, exacerbate retinal and macular edema. In contrast, the full netrin-1 protein maintains blood–brain barrier function and mitigates the development of diabetic retinopathy [7]. 

### 3.6. Chronic Inflammation of Lungs 

Patients with idiopathic lung fibrosis showed high expression levels of netrin-1. Therefore, netrin-1, which interacts with its receptor DCC, promotes lung fibrosis with histopathological changes and remodels adrenergic nerves [149]. 

Kerget et al. [150] found that patients with an acute exacerbation of COPD showed a higher expression of netrin-1 than a control group. More studies are necessary to evaluate these findings.

## 4. Discussion

Even though much has been learned over the last two decades since netrin-1 was first described outside the central nervous system, many new questions have emerged. 

Collectively, most studies show that the regulation of netrin-1 is critically involved in acute (Figure 1) and chronic inflammation (Figure 2). The expression of netrin-1 and its various receptors are differentially regulated depending on the tissue, the stimulus, the source, and the target cell (Table 1 and Table 2). Although the effects of netrin-1 in acute and chronic inflammation seem to be opposed, most studies show that netrin-1 can effectively change the kinetics and functions of the leukocytes in the setting of inflammation.

In chronic inflammation, adipose tissue macrophages and plaque macrophages secrete netrin-1, which actively hinders the egress of macrophages out of the inflamed tissue. Changing the local proinflammatory environment, however, is key for overcoming the chronic ongoing inflammation. Specifically targeting these persistent proinflammatory macrophages through the secretion of netrin-1 and, in this context, its main receptor Unc5b is a very promising target in order to beneficially modify the local environment, overcome the ongoing inflammation, promote resolution, and restore homeostasis. In chronic inflammation, cells of the adaptive immune system play a key role. After being challenged by antigen-presenting cells such as monocytes, macrophages, and dendritic cells, B- and T-cells are activated and recruited into the inflamed tissue. Understanding how netrin-1 either directly or indirectly, through the polarization of monocytes, affects the adaptive immune response in chronic inflammation will be key to further delineating the effects by which netrin-1 promotes and sustains chronic inflammation.

In acute inflammation, the systemic application has been shown to mainly mitigate the acute inflammatory response and mitigate leukocyte influx into the inflamed tissue predominantly involving the purinergic pathway through the adenosine A2B receptor. In the early phase of acute inflammation, involving primarily the innate immune system, netrin-1 is further able to promote the infiltrating monocytes towards an anti-inflammatory and pro-resolving phenotype. However, all the models of acute inflammation that have been studied do not involve any exogenous pathogens, which must be controlled. These models all have an impaired restraint of the innate immune system in common, in which netrin-1 effectively mitigates the onset of the inflammation and in the latter promotes the resolution. However, it will be important to understand whether, in the case of an underlying pathogen, this extenuated immune response is still able to overcome the invading pathogen or whether it contributes to fatal outcomes.

The great challenge in the field remains understanding how netrin-1 is regulated under various conditions. Further, the downstream mechanisms and pathways of its various receptors need further study to understand how netrin-1 regulates the organ-specific activation and differentiation of tissue-resident and hematopoietic leukocytes under distinct acute and chronic inflammatory conditions.

## Figures and Tables

**Figure 1 ijms-23-00275-f001:**
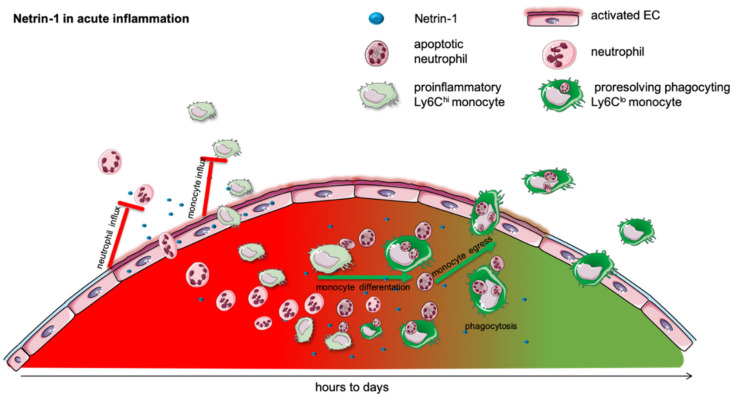
Netrin-1 in acute inflammation. Acute inflammation is defined by a rapid influx of neutrophils into the inflamed tissue to combat the intruding pathogen. This is followed by a monocyte recruitment to extend the rapid immune response. In many different models, it has been shown that netrin-1 can very effectively mitigate the influx of neutrophils and monocytes and thereby dampen the acute inflammatory response. The acute, initial phase is followed by a resolving phase, in which neutrophils undergo apoptosis and are taken up by monocytes. The efferocytosis in turn promotes the differentiation of pro-inflammatory into anti-inflammatory monocytes. This phenotype switch and the monocyte egress in acute inflammation is facilitated by netrin-1.

**Figure 2 ijms-23-00275-f002:**
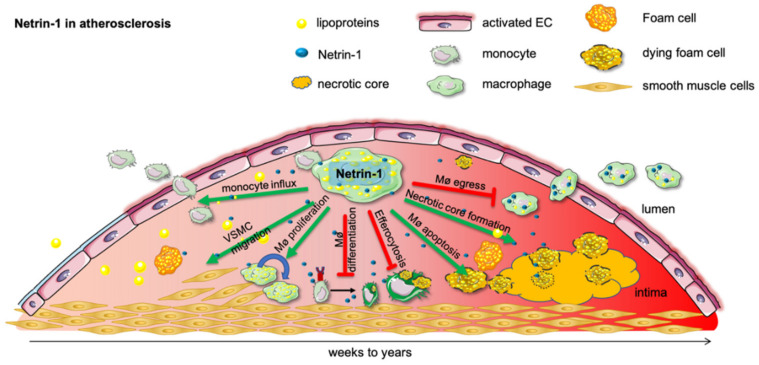
Netrin-1 in atherosclerosis. Subendothelial lipoprotein accumulation incite a proinflammatory activation of the endothelial cells and promote monocyte influx into the developing atherosclerotic plaque. As lipoproteins undergo various modifications, such as oxidation and hydrolysis, they are taken up by monocytes and form proinflammatory foam cells. In this inflammatory milieu chronically, activated macrophages secrete netrin-1, which recruit further monocytes and retain macrophages in the tissue until they undergo apoptosis and form the detrimental necrotic core. Netrin-1 further mitigates pro-resolving macrophage differentiation and efferocytosis. Moreover, netrin-1-secreting macrophages advance the plaque further by recruiting smooth muscle cells into the developing plaque, which can themselves become macrophages and promote the ongoing progression of the plaque.

**Table 1 ijms-23-00275-t001:** Murine studies on netrin-1 and its receptors in acute inflammation.

Acute Inflammation				
Model	Receptor	Netrin-1 Expression	Outcome	Ref.
Lung Inflammation				
*S. aureus* pneumonia in mice	Unc5b	⇓	Netrin-1 expression is attenuated in *S. aureus* septicemia.	[20]
LPS and ventilator induced lung injury in mice	Adora2b	⇓	Netrin-1 expression is attenuated by inflammation, and netrin-1 limits neutrophil influx into the lung.	[29]
LPS induced lung injury in pigs		not defined	Intravenous and inhalative netrin-1 mitigated pulmonary inflammation and lung damage.	[30]
Mouse model with LIRI	Adora2b	⇓	Positive correlation between netrin-1 expression and Treg cell population.	[31]
An ALI model was established by intratracheal instillation of LPS in C57BL/J mice	Adora2b	⇓	Receptor binding has the potential to enhance ENaC-dependent alveolar fluid clearance by supplementation of netrin-1.	[32]
Hypobaric hypoxia-induced lung injury in mice	UNC5HB	⇓	Pretreatment of netrin-1 dampen ALI and inhibits neutrophil migration.	[33]
Abdominal Inflammation				
Colitis				
DSS-induced colitis	Adora2b	⇑ in colon	Netrin-1 is induced during acute colitis limits neutrophil influx into the colonic epithelium.	[34]
Peritonitis				
ZyA-induced peritonitis	Adora2b	⇓	Netrin-1 expression is reduced during peritonitis. Exogenous netrin-1 application attenuates inflammation.	[35]
ZyA-induced peritonitis	Adora2b		Netrin-1 synergistically interacts with RvD2 in an anti-inflammatory and pro-resolving fashion.	[27]
Pancreatitis				
L-arginine-induced acute pancreatitis	not defined	⇓	Netrin-1 administration reduced pancreas and lung injury.	[36]
Acute Kidney Injury				
Ischemia–reperfusion-induced AKI	not defined	⇑ protein levels in renal tissue mRNA in renal tissue protein and mRNA levels in intestine	Netrin-1 is induced in renal tissue following IR. Ntn1 administration attenuates renal failure and kidney inflammation.	[37]
Ischemia–reperfusion-induced AKI	not defined	not defined	Netrin-1 overexpression attenuates renal failure by decreased apoptosis and increased tubular epithelium proliferation.	[38]
Ischemia–reperfusion-induced AKI	Unc5b	⇓ mRNA in renal tissue	Netrin-1 overexpression attenuates renal failure and systemic inflammation.	[39]
Ischemia–reperfusion-induced AKI	not defined	⇑ renal tissue	Netrin-1 attenuates renal failure and systemic inflammation.	[40]
Ischemia–reperfusion-induced AKI	not defined	not defined	Netrin-1 attenuates renal failure and promotes M2 polarization through PPARγ.	[16]
Hypoxia				
In vivo hypoxia	Adora2b	⇑ in lung and colon via HIF-1α	Netrin-1 attenuates systemic inflammation and neutrophil recruitment.	[13]
Liver-Ischemia Reperfusion				
	Adora2b	⇓ in liver	Netrin-1 attenuates neutrophil recruitment, local and systemic inflammation and promotes resolution and tissue regeneration.	[28]
Myocardial Infarct				
Normothermic ischemia reperfusion Langendorff perfusion	DCC	not defined	Netrin-1 pre- and postconditioning decrease infarct size and attenuate myocardiocyte apoptosis through ERK1/2 and NO induction.	[41,42,43]
Heterotopic cardiac transplant with 8 h of cold ischemia		⇓	Netrin-1 protects from reperfusion injury by limiting leukocyte influx, cardiomyocyte apoptosis, and M2 macrophage polarization through PPARg.	[44]
Chronic LAD coronary ligation	not defined	not defined	Intracardial application of netrin-1-transduced mesenchymal stem cells attenuated infarction size and prevented cardiac hypertrophic remodeling.	[45]
Normothermic ischemia reperfusion Langendorff perfusion	not defined	not defined	Netrin-1 preconditioning attenuates infarcts size by preventing superoxide and NADPH oxidase production, preventing mitochondrial dysfunction.	[46]

**Table 2 ijms-23-00275-t002:** In vitro and in vivo studies on netrin-1 and mechanisms in chronic inflammation.

Chronic Inflammation				
Model	Receptor	Netrin-1 Expression	Outcome	Ref.
Atherosclerosis				
Analysis of p.R590L variant of netrin-1 (mutNetrin-1)	UNC5B DCC Neogenin		Decrease in binding capacity to UNC5B and DCC and an increase in binding capacity to neogenin. Stimulation of monocyte adhesion and expression of IL-6, CCL2, and ICAM-1. Diminishing macrophages and smooth muscle cell migration.	[97]
Induction of Raw264.7 macrophages with oxLDL	UNC5B	⇑	Downregulation of CCR7 expression and inhibition of macrophage migration.	[98]
Bone marrow transplantation of Ntn-1−/−-deficient cells in LDLR−/− KO mice on western diet in plaque progression	UNC5B Neogenin	⇑	Hematopoetic netrin-1 KO prevents atherosclerosis development by mitigating MØ and VSMC influx and facilitating MØ egress	[99]
Monocyte- and macrophage-specific tamoxifen-inducible CX3CR1-driven cre recombinase netrin-1 floxed mice (Ntn1fl/fl Cx3cr1creERT2+) in plaque regression	UNC5B		Reduced plaque size and complexity in aortic wall, inflammation resolution, IL-10 production, and efferocytosis by myeloid Ntn1 deletion.	[100]
Obesity				
Mouse model of diet-induced obesity	Unc5B	⇑ in obese, but not lean adipose tissue	Expression of netrin-1 and its receptor are regulated by saturated fatty acid. Macrophages with a reduced migratory capacity. Restored by blocking netrin-1.	[52]
Hematopoietic deletion of Ntn1		⇓	Relief of adipose tissue macrophage emigration, reduction of inflammation, and improvement of insulin sensitivity.	[52]
Mouse model with myeloid- specific deletion of netrin-1 (Ntn1fl/fl LysMCre+/−; Ntn1 mac)	not defined		Ntn1 mac mice: modest decrease in HFD-induced adiposity and adipocyte size. Ntn1 mac macrophages: reduced expression of genes involved in arachidonic acid metabolism and decreases in proinflammatory eicosanoids. Myeloid-specific deletion of netrin-1 caused a decrease of ATMs, particularly the resident macrophage subset. Macrophages reprogram the ATM phenotype, leading to reduced adipose inflammation and improved lipid handling and metabolic function.	[101]
Diabetes				
Adipose-derived stem cells modified by netrin-1 gene (NTN-1) in vitro, condition of high glucose		⇑	Proliferation, migration, adhesion, and inhibition of the apoptosis of ADSCs.	[102]
Injected adipose-derived stem cells modified by netrin-1 gene (NTN-1) in vivo (sciatic denervated mice with type 2 diabetes mellitus)		⇑	Capillaries and endothelium were formed by differentiation of N-ADSCs, higher density of microvessels. Upregulation of AKT/PI3K/eNOS/P-38/NF-κB signaling pathways.	[102]
Exogenous netrin-1 in UNC5B-depleted human renal glomerular endothelial cells (HRGECs)	UNC5B	not defined	Inhibition of cell migration and tubulogenesis. Association with SRC pathway deactivation UNC5B antagonizes netrin-1 and that UNC5B upregulation for the enhancement of angiogenesis.	[103]
Destructive Joint Disease / Osteoarthritis				
Female mice with or without knockout of netrin-1 or receptor to detect differences in expression and effect on bone structure.	UNC5B	⇑	Activation of SHP1, inhibition of multinucleation of osteoclasts and preventing bone erosion in autoimmune arthritis.	[104]
Mouse model with implanted ultrahigh-molecular-weight-polyethylene particles (UHMWPE) over the calvaria and weekly injection of antibodies for netrin-1 and its receptor	UNC5B	not defined	Reduced particle-induced bone pitting, inflammatory processes, and TRAP (tartrate-resistant acid phosphatase)-positive osteoclasts.	[105]
RAW 264.7 mouse monocyte macrophages and air pouch model of bone resorption	UNC5B	not defined	Effect of netrin-1 via the ERK1/2 signaling pathway on osteoclast development by promoting autophagy.	[106]

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
