# Peer review of "Netrin-1: A Modulator of Macrophage Driven Acute and Chronic Inflammation"

_ijms, 2021, doi:10.3390/ijms23010275_

Round 1
Reviewer 1 Report
In this review article, the authors aimed to highlight the dichotomous roles of netrin-1 and its interaction with monocytes and macrophages and their impact on acute and chronic inflammation.
However, in this descriptive review on netrin involvement in a lot of pathologies associated either with acute or chronic inflammation, the macrophage involvement is not always presented. Therefore, I suggest to the authors, either to change the title of the review, or present in the review only the studies were macrophages and netrin are associated.
Other suggestions to the authors:
- to give titles to to Fig.1 and Fig.2;
-to correct editing errors;
-to place section 4(Acute inflammation) before section 3 (Chronic inflammation)
Author Response
Response to Reviewer 1 Comments
- However, in this descriptive review on netrin involvement in a lot of pathologies associated either with acute or chronic inflammation, the macrophage involvement is not always presented. Therefore, I suggest to the authors, either to change the title of the review, or present in the review only the studies were macrophages and netrin are associated.
Thank you for the very valid comment. As during writing the scope had extended, we have changed the title to “Netrin-1: A modulator of acute and chronic inflammation”. Consequently, we have removed the first Paragraph on macrophages and reorganized the introduction.
- Give titles to to Fig.1 and Fig.2:
The figure titles have now been incorporated into the figure legend.
- Correct editing errors:
We have extensively corrected editing errors.
- Place section 4(Acute inflammation) before section 3 (Chronic inflammation):
Following your suggestions, the sections have been swapped.
Thanks again for your precise and constructive comments
Reviewer 2 Report
This review is extensive and well researched however the English needs to be improved with good proofreading (see attached document for suggestions) and the organisation of the tables and figures should be changed. The figures should be referred to in the text and the tables should be placed near to the text that refers to them.
When a paragraph starts with a reference the authors do not add the reference number until the end of the section, ie in line 173 the authors wrote; In this context, Yang et al. focused on the expression of netrin-1... but the reference was in line 177. This should be avoided and the reference number should be included after the Yang et al..

Author Response
Response to Reviewer 2 Comments
- This review is extensive and well researched however the English needs to be improved with good proofreading (see attached document for suggestions) and the organisation of the tables and figures should be changed.
Thank you for your extensive proof reading. In addition to your proofs, we have had the manuscript edited by the IJMS English editing service.
- The figures should be referred to in the text and the tables should be placed near to the text that refers to them.
We have moved the tables and figures next to the referring text now.
- When a paragraph starts with a reference the authors do not add the reference number until the end of the section, ie in line 173 the authors wrote; In this context, Yang et al. focused on the expression of netrin-1... but the reference was in line 177. This should be avoided and the reference number should be included after the Yang et al..
Thanks for this valuable remark. We have updated the text and referencing accordingly.
Round 2
Reviewer 1 Report
The manuscript has been
sufficiently improved to warrant publication in IJMS.
Author Response
A spell check and language recheck has been performed.
Please find the minor changes in the tracked document.
Reviewer 2 Report
The paper looks better since the first review, however the two tables are not referred to in the text. I think this is a serious omission.
Author Response
We are referring to the tables now in the first paragraph of acute and chronic inflammation respectively and in the discussion.
A spell check and language recheck has been performed.
Please find the minor changes in the tracked document.